# Epigenetic Biomarkers in Thrombophilia-Related Pregnancy Complications: Mechanisms, Diagnostic Potential, and Therapeutic Implications: A Narrative Review

**DOI:** 10.3390/ijms252413634

**Published:** 2024-12-20

**Authors:** Claudia Ramona Bardan, Ioana Ioniță, Maria Iordache, Despina Călămar-Popovici, Violeta Todorescu, Roxana Popescu, Brenda Cristiana Bernad, Răzvan Bardan, Elena Silvia Bernad

**Affiliations:** 1Doctoral School, Faculty of Medicine, “Victor Babeș” University of Medicine and Pharmacy, 300041 Timisoara, Romania; claudia.iusco@umft.ro (C.R.B.); bernad.brenda@umft.ro (B.C.B.); 2Clinic of Hematology, Municipal Clinical Emergency Hospital, 300254 Timisoara, Romania; ionita.ioana@umft.ro (I.I.); iordache.maria@umft.ro (M.I.); calamar.despina@umft.ro (D.C.-P.); todorescu.violeta@umft.ro (V.T.); 3Department of Hematology, “Victor Babeș” University of Medicine and Pharmacy, 300041 Timisoara, Romania; 4Division of Cell and Molecular Biology, Department of Microscopic Morphology, “Victor Babeș” University of Medicine and Pharmacy, 300041 Timisoara, Romania; popescu.roxana@umft.ro; 5Center for Neuropsychology and Behavioral Medicine, “Victor Babeș” University of Medicine and Pharmacy, 300041 Timisoara, Romania; 6Department of Urology, “Victor Babeș” University of Medicine and Pharmacy, 300041 Timisoara, Romania; 7Clinic of Urology, “Pius Brînzeu” County Clinical Emergency Hospital, 300723 Timisoara, Romania; 8Department of Obstetrics and Gynecology, “Victor Babeș” University of Medicine and Pharmacy, 300041 Timisoara, Romania; bernad.elena@umft.ro; 9Clinic of Obstetrics and Gynecology, “Pius Brînzeu” County Clinical Emergency Hospital, 300723 Timisoara, Romania; 10Center for Laparoscopy, Laparoscopic Surgery and In Vitro Fertilization, “Victor Babeș” University of Medicine and Pharmacy, 300041 Timisoara, Romania

**Keywords:** thrombophilia, pregnancy, epigenetic biomarkers, miRNA, lncRNA, extracellular vesicles

## Abstract

Pregnancy complications associated with thrombophilia represent significant risks for maternal and fetal health, leading to adverse outcomes such as pre-eclampsia, recurrent pregnancy loss, and intra-uterine growth restriction (IUGR). They are caused by disruptions in key physiological processes, including the coagulation cascade, trophoblast invasion, angiogenesis, and immune control. Recent advancements in epigenetics have revealed that non-coding RNAs, especially microRNAs (miRNAs), long non-coding RNAs (lncRNAs), and extracellular vesicles (EVs) carrying these RNAs, play crucial roles in the regulation of these biological processes. This review aims to identify the epigenetic biomarkers that are the best candidates for evaluating thrombophilia-related pregnancy complications and for assessing the efficacy of anticoagulant and antiaggregant therapies. We emphasize their potential integration into personalized treatment plans, aiming to improve the risk assessment and therapy strategies for thrombophilic pregnancies. Future research should focus on validating these epigenetic biomarkers and establishing standardized protocols to enable their integration into clinical practice, paving the way for a precision medicine approach in obstetric care.

## 1. Introduction

Pregnancy is a complex and dynamic process requiring sophisticated physiological regulation mechanisms to ensure successful maternal and fetal outcomes [1]. The hemostatic system has a key role in these regulation processes, maintaining the optimal balance between procoagulant and anticoagulant mechanisms in order to ensure normal placental development, trophoblast invasion, angiogenesis, and immune modulation [2]. Any disturbance in this system can lead to significant adverse pregnancy outcomes, including pre-eclampsia, pregnancy loss, intra-uterine growth restriction (IUGR), and venous thromboembolism. At this moment, an accumulating body of evidence suggests that thrombophilias (congenital and acquired), along with other coagulation disorders, have a significant contribution to the mentioned pregnancy complications [3].

Congenital thrombophilias, which include Factor V Leiden mutation and prothrombin gene mutation, are genetic predispositions leading to hypercoagulable states and increasing the risk of thromboembolic events [4]. Women experiencing recurrent pregnancy loss or pregnancy complications like pre-eclampsia frequently receive a diagnosis of acquired thrombophilias, primarily due to the antiphospholipid syndrome (APS) [5]. They can disrupt the homeostasis of the coagulation cascade, which is essential for utero-placental circulation and for fetal development. This disruption may result in placental thrombosis, impaired trophoblast invasion, or abnormal angiogenesis, with consecutive negative pregnancy outcomes [6].

Recent research has concentrated on finding and understanding the molecular and epigenetic regulators of the processes underlying the coagulation cascade. Epigenetic processes, including microRNAs (miRNAs), long non-coding RNAs (lncRNAs), and extracellular vesicles (EVs), have been identified as key elements regulating gene expression associated with coagulation pathways and thrombophilia [7].

Taking into account the significant role of these epigenetic regulators in the pathophysiology of thrombophilias and other coagulation disorders during pregnancy, there is a growing interest in identifying specific biomarkers to assist in the diagnosis and management of these conditions [8]. Furthermore, miRNAs could serve as novel tools to track the effectiveness of antiaggregant and anticoagulant therapies, commonly prescribed to mitigate thrombophilia risks during pregnancy [9].

The main objective of this review is to explore the fundamental pathophysiological mechanisms of congenital and acquired thrombophilias, along with other coagulation abnormalities during pregnancy, focusing on their effects on trophoblast invasion, angiogenesis, and immunological modulation. Additionally, we will emphasize the potential of miRNAs, lncRNAs, and extracellular vesicles as biomarkers for these conditions and their potential utility in assessing the efficacy of antiaggregant and anticoagulant therapies to improve pregnancy outcomes. Integrating our current knowledge of these mechanisms and biomarkers, our ultimate goal is to identify novel diagnostic and therapeutic options that can improve maternal and fetal health in thrombophilic patients.

## 2. Physiology of the Coagulation System During Pregnancy

### 2.1. Adaptations of the Coagulation Cascade During Pregnancy

The coagulation system consists of a complex sequence of enzymatic reactions that lead to the formation of fibrin, the essential structural component of blood clots [10]. During the normal hemostasis process, the coagulation cascade is activated after vascular injury to prevent hemorrhage by creating a clot that effectively seals the injured vessel [11].

During the entire period of pregnancy, the maternal organism experiences a state of hypercoagulability, which is considered an adaptive response mechanism to protect the mother from uncontrolled hemorrhage during childbirth. This hypercoagulable state is characterized by elevations in the levels of clotting factors (including fibrinogen, Factor VII, and Factor VIII), reduced fibrinolysis, and increased platelet activity [12]. Although this physiological shift is beneficial, it also increases the risk of thromboembolic events, especially in women with predisposing conditions, including underlying thrombophilias [13,14,15,16].

### 2.2. Role of Coagulation in Trophoblast Formation and Invasion

Trophoblast formation and invasion is an essential stage in the initial phases of pregnancy. The trophoblasts are specialized cells originating from the outer layer of the blastocyst that have an essential role in the process of embryonic implantation and in the formation of the placenta [17]. The trophoblast has two layers: the outer layer is composed of syncytiotrophoblasts and is responsible for nutrient exchange, while the inner layer with cytotrophoblasts invades the maternal endometrium, establishing a blood supply for the developing embryo [18].

Trophoblast invasion is a highly regulated process requiring the breakdown of the maternal endometrial extracellular matrix and the remodeling of maternal spiral arteries, in order to guarantee an adequate blood supply to the placenta [19]. Coagulation factors, especially thrombin, play a dual role during this process. Initially, thrombin facilitates trophoblast invasion by activating proteases such as matrix metalloproteinases (MMPs), which degrade the extracellular matrix [20]. However, excessive thrombin formation (observed in thrombophilic patients) can lead to premature fibrin deposition around the spiral arteries, restricting trophoblast invasion and impairing placental blood flow [21].

The role of excessive coagulation in impairing trophoblast function was confirmed in several studies [22,23]. For instance, mutations in genes that encode coagulation proteins can lead to increased fibrin accumulation in the placental intervillous spaces, disrupting nutrient exchange and leading to placental insufficiency [24].

### 2.3. Angiogenesis and the Coagulation Cascade

Angiogenesis is a vital process of pregnancy, especially during placental development, as optimal vascularization of the placenta guarantees that the fetus receives an adequate intake of oxygen and nutrients during gestation [25]. The regulation of angiogenesis implicates the maintaining of a delicate balance between pro-angiogenic factors (including vascular endothelial growth factor—VEGF, and placental growth factor—PlGF) and anti-angiogenic factors (like soluble fms-like tyrosine kinase-1, which acts as a VEGF antagonist) [26].

The coagulation system interacts with angiogenesis at multiple levels. Thrombin, considered an essential mediator of coagulation, can promote angiogenesis by inducing endothelial cell proliferation and the expression of pro-angiogenic proteins such as VEGF. However, in a thrombophilic state, increased thrombin production can paradoxically inhibit angiogenesis. Research indicates that high levels of thrombin can activate endothelial cell receptors (including protease-activated receptor-1—PAR-1), leading to the release of anti-angiogenic factors and facilitating endothelial dysfunction [27]. Such an imbalance between pro- and anti-angiogenic signals is characteristic of pregnancy complications such as pre-eclampsia and IUGR, both associated with inadequate placental vascularization. Moreover, increased fibrin deposition in the placenta vasculature (found in pregnant female patients with Factor V Leiden mutation and in antiphospholipid syndrome) leads to reduced blood flow and oxygenation, impairing angiogenesis, and finally reducing placental vascularization [28].

### 2.4. Immune Regulation and Coagulation

The maternal immune system has to tolerate the semi-allogeneic fetus while preserving its capacity to defend against infections. This balance, which is essential for a successful pregnancy, is also influenced by the coagulation system: immune cells, including macrophages, natural killer (NK) cells, and T lymphocytes, interact with trophoblasts and placental endothelial cells, modulating the maternal–fetal interface [29,30].

Thrombin activates the immune cells by attaching on their surface to protease-activated receptors (PARs), releasing pro-inflammatory cytokines. During normal pregnancy, the inflammatory signals are carefully regulated to reduce excessive inflammation. However, in thrombophilic conditions, hypercoagulation can trigger an exaggerated immune response, leading to placental inflammation and damage [31].

The coagulation system also has a role in regulating the balance between Th1 and Th2 immune responses during pregnancy: a Th1-dominant response is correlated with increased inflammation and has a negative role in pregnancy, while a Th2-dominant response promotes immune tolerance and is important for fetal survival [32]. The pro-inflammatory effect of the increased levels of thrombin in pregnant women with clinically manifest thrombophilias can disrupt the balance, favoring the Th1-dominant response, and contributing to the development of pre-eclampsia [33].

## 3. The Impact of Thrombophilias During Pregnancy

### 3.1. Complications of Thrombophilias During Pregnancy

#### 3.1.1. Intra-Uterine Growth Restriction (IUGR)

IUGR occurs due to inadequate placental perfusion, often driven by thrombophilia-induced vascular insufficiency. Thrombi in the placental vasculature reduces the delivery of oxygen and nutrients to the fetus, impairing growth and development [34].

#### 3.1.2. Recurrent Pregnancy Loss (RPL)

Recurrent pregnancy loss (RPL) is a common complication of thrombophilias. In these conditions, excessive clot formation within placental vessels leads to microthrombosis, disrupting trophoblast function and nutrient delivery to the embryo [35].

#### 3.1.3. Stillbirth

Thrombophilias are a leading cause of stillbirth due to their impact on placental function, predisposing to placental infarction, chronic hypoxia, and fetal death. In cases of unexplained stillbirth, thrombophilias should be considered as underlying etiological factors, especially when other risk factors are absent [36].

#### 3.1.4. Pre-Eclampsia

This condition, characterized by hypertension, proteinuria, and end-organ dysfunction, is associated with thrombophilic states. The prothrombotic environment associated with thrombophilias exacerbates maternal vascular damage, increasing the likelihood of severe complications, including the HELLP syndrome [37].

#### 3.1.5. Venous Thromboembolism (VTE)

Venous thromboembolism is among the most severe complications of thrombophilia during pregnancy. Deep vein thrombosis (DVT) often presents as unilateral leg swelling, pain, and erythema. It is also concerning in the postpartum period, when the risk of VTE is highest due to immobility, endothelial injury, and sustained hypercoagulability [38]. Pulmonary embolism (PE), a potentially fatal complication of VTE, occurs when a thrombus dislodges and travels to the pulmonary arteries, causing chest pain, dyspnea, and hypoxia [39]. In thrombophilic pregnancies, the risk of PE increases exponentially, requiring careful screening and prophylactic anticoagulation in high-risk patients [40].

### 3.2. Congenital Thrombophilias in Pregnancy

#### 3.2.1. Factor V Leiden Mutation

Factor V Leiden mutation represents the most common congenital thrombophilia, affecting 3–8% of the Caucasian population, making the Factor V protein resistant to inactivation by activated protein C (APC). Therefore, Factor V remains active longer than normal, generating an excessive production of thrombin and increasing the risk of thrombosis [15,41]. During pregnancy, the mutation has been associated with an increased risk of recurrent miscarriage, especially in the second and third trimesters. Most probably it causes an excessive accumulation of fibrin in the placental vasculature, resulting in placental insufficiency and impaired nutrient exchange between the mother and the fetus. However, not all women with the mutation experience complications, suggesting that other genetic or environmental factors may influence the outcome [42].

#### 3.2.2. Prothrombin Gene Mutation (G20210A)

The prothrombin gene mutation G20210A occurs in approximately 2–3% of the general population, leading to elevated levels of prothrombin (Factor II), with an increased risk of venous thromboembolism [43]. Elevated prothrombin levels during pregnancy increase the risk of abnormal clot formation in the placental vasculature, similar to the mechanisms observed in the Factor V Leiden mutation [5].

Research indicates that women carrying the prothrombin mutation are at increased risk for late pregnancy complications, such as IUGR, pre-eclampsia, and placental abruption, due to placental ischemia and infarctions [44].

#### 3.2.3. Deficiencies in Natural Anticoagulants

Women with protein C, protein S, or antithrombin deficiencies are at increased risk of recurrent pregnancy loss, IUGR, and pre-eclampsia, especially during the second and third trimesters. As these proteins are considered essential in regulating the coagulation cascade by inhibiting thrombin production and promoting fibrinolysis, their deficiencies can lead to uncontrolled coagulation activity, finally increasing the risk of placental thrombosis and associated complications [45].

#### 3.2.4. MTHFR Mutation

The MTHFR gene encodes the enzyme methylenetetrahydrofolate reductase, which regulates the folate metabolism and the levels of homocysteine., which are involved in folate metabolism and homocysteine regulation. Mutations in this gene, especially the C677T variant, are associated with elevated homocysteine levels (hyperhomocysteinemia), a condition associated with an increased risk of thrombosis [46]. Although the role of MTHFR mutations in pregnancy complications remains controversial, some studies have suggested an association between hyperhomocysteinemia and recurrent pregnancy loss, pre-eclampsia, and IUGR, caused by endothelial dysfunction, oxidative stress, and increased platelet activation [47,48].

### 3.3. Acquired Thrombophilias in Pregnancy

#### Antiphospholipid Syndrome (APS)

APS is characterized by the presence of antiphospholipid antibodies (aPLs), including lupus anticoagulant, anticardiolipin antibodies, and anti-β2 glycoprotein I antibodies, which target phospholipid-binding proteins on endothelial cells, platelets, and trophoblasts, promoting coagulation and inflammation [49]. Studies have shown that women with APS are at high risk of miscarriage, especially during the second and third trimesters, which is primarily due to placental thrombosis, resulting in placental insufficiency and fetal hypoxia [16,49]. Furthermore, APS can impair trophoblast invasion and angiogenesis, processes that are critical for the development of an adequate utero-placental blood supply, leading to pre-eclampsia and IUGR [50].

In addition to promoting thrombosis, antiphospholipid antibodies contribute to pregnancy complications by inducing a pro-inflammatory environment at the maternal–fetal interface, shifting the balance toward a Th1-dominant response, and further increasing the risk of pregnancy complications (Figure 1) [51,52].

Table 1 describes the impact of the different components of the coagulation system on the physiological processes of pregnancy, considering both normal development and pathological development due to thrombophilias.

## 4. Epigenetic Regulation in Coagulation During Pregnancy

Recent research has identified a critical role for epigenetic mechanisms, particularly through non-coding RNAs such as microRNAs (miRNAs) and long non-coding RNAs (lncRNAs), in regulating the coagulation system during pregnancy. These epigenetic regulators can influence the expression of genes involved in the coagulation cascade, angiogenesis, and immune responses, thereby modulating the risk of thrombophilia-related complications [53]. We concentrate on the epigenetic factors related to the maternal side, as our main focus is on maternal coagulation problems and complications.

### 4.1. Role of miRNAs in Coagulation and Vascular Health

MicroRNAs are small non-coding RNAs that regulate gene expression by binding to messenger RNA (mRNA) targets, inhibiting translation, or promoting mRNA degradation. In pregnancy, miRNAs are particularly effective in regulating genes involved in coagulation, endothelial function, and immune tolerance, making them essential in managing hemostasis [54].

The role of several miRNAs was already determined in modulating the expression of key coagulation factors that influence the extrinsic and intrinsic pathways, controlling thrombin formation and fibrinolysis [55].

For instance, miR-223 has been shown to regulate the expression of tissue factor, a protein that initiates the extrinsic pathway of coagulation by activating Factor VII [56]. In thrombophilic patients, dysregulation of miR-223 can result in increased tissue factor expression, thereby promoting thrombin generation and clot formation [57]. Recent studies have shown that miR-223-3p is downregulated in the placental tissues of patients with preeclampsia, increasing the secretion of inflammatory cytokines such as IL-1β and IL-18 [58].

Similarly, miR-146a is known to suppress the expression of PAI-1, a key inhibitor of fibrinolysis [59]. In conditions where miR-146a is downregulated, PAI-1 levels increase, leading to impaired clot breakdown and enhanced thrombosis risk [60]. Recent studies have shown that patients with aPLs and pregnancy complications expressed significantly higher levels of circulating miR-146a-3p. Upregulation of miR-146a-3p played a major role in the development of obstetric APS, causing trophoblast cells to release interleukin-8 by activating TLR8 [61].

Multiple studies have confirmed that miR-210, which is highly expressed in hypoxic conditions, is involved in placental development and angiogenesis. Dysregulation of miR-210 inhibits cytotrophoblastic cell migration in the myometrial portions of the spiral arteries, causing impaired placental perfusion and excessive coagulation and finally leading to pre-eclampsia and IUGR. In pathological states, elevated miR-210 levels can lead to endothelial dysfunction, indirectly influencing the levels or activity of Factor V, fibrinogen, or other coagulation proteins (Figure 2) [62,63].

Moreover, miR-126, a well-known regulator of endothelial function and angiogenesis, is critical in maintaining vascular integrity during pregnancy, restricting leukocyte adhesion to the endothelium, and constraining TF gene expression [64]. Alterations in miR-126 levels have been implicated mostly in pre-eclampsia, where low levels of vascular endothelial growth factor (VEGF) as a sign of endothelial dysfunction were found, and less in recurrent pregnancy loss [65].

MiR-145 has been identified as a key factor for regulating thrombus formation in venous thromboembolism (VTE): levels of miR-145 showed an inverse correlation with thrombus load, while miRNA target prediction tools have identified tissue factor (TF) as a genetic target for miR-145. Further studies will need to find eventual relationships between thrombophilias and the early detection of VTE during pregnancy [66].

### 4.2. Role of lncRNAs in Coagulation and Immune Modulation

LncRNAs are a class of non-coding RNAs exceeding 200 nucleotides that regulate gene expression through a variety of mechanisms, including chromatin remodeling, interference with transcription factors, and post-transcriptional control. Recent studies have demonstrated the significance of lncRNAs in the regulation of the coagulation system, particularly in the context of endothelial function, platelet activity, and immunological responses, all of which are essential for sustaining normal pregnancy [67,68].

Endothelial cells, which make up the lining of the blood vessels, are essential in modulating coagulation and vascular function. LncRNAs regulate endothelial cell activity by influencing the expression of genes associated with angiogenesis, inflammation, and coagulation [68]. For instance, lncRNA MALAT1 (metastasis-associated lung adenocarcinoma transcript 1) has been demonstrated to enhance endothelial cell proliferation and migration through the regulation of VEGF production, a key driver of angiogenesis [69]. Dysregulation of MALAT1 has been associated with endothelial dysfunction in conditions like pre-eclampsia, where abnormal angiogenesis and excessive coagulation contribute to placental insufficiency [70,71,72].

Another lncRNA, TUG1 (taurine upregulated gene 1), has been involved in endothelial cell survival and apoptosis [73]. Studies have shown that TUG1 levels are modified in women with pre-eclampsia, suggesting that this lncRNA may play a role in endothelial dysfunction [74]. TUG1 may also regulate the expression of coagulation factors such as Factor V and Factor VIII, thereby influencing thrombin generation and fibrin formation [75].

In addition to their role in endothelial function, lncRNAs regulate immune responses during pregnancy, especially at the level of the maternal–fetal interface [76]. LncRNA HOTAIR (HOX transcript antisense RNA) regulates pro-inflammatory cytokine expression in immune cells, hence influencing the balance between pro- and anti-inflammatory signals [77,78]. In thrombophilic patients, excessive inflammation can initiate coagulation via the activation of tissue factors and the release of pro-thrombotic cytokines. Dysregulation of HOTAIR could aggravate the inflammatory state seen in disorders like APS, where immunological activation leads to placental thrombosis and even pregnancy loss [79].

### 4.3. Extracellular Vesicles and Their Role in Coagulation and Pregnancy

Extracellular vesicles (EVs), which include exosomes and microvesicles, are membrane-bound particles released by various cell types, including trophoblasts, endothelial cells, and immune cells [80]. EVs may transport proteins, lipids, and nucleic acids, including miRNAs and lncRNAs, which can affect cellular activities in adjacent and remote cells. By transferring miRNAs and lncRNAs to target cells, EVs are able to influence gene expression, coagulation, and immune responses, making them key players in thrombophilia-related pregnancy complications [81].

EVs originating from trophoblasts and endothelial cells are known to carry pro-coagulant proteins, such as tissue factor, which activates the extrinsic pathway of coagulation [82]. In pregnant women with thrombophilia, an increased concentration of tissue factor-bearing EVs in the maternal circulation promotes a hypercoagulable state, contributing to placental thrombosis and vascular insufficiency [83]. In addition, EVs carrying specific miRNAs and lncRNAs can modulate endothelial cell function, impacting vascular health and the risk of placental hypoxia [84,85,86]. They can also deliver pro-coagulant molecules such as TF, phosphatidylserine, or other coagulation factors to the maternal–fetal interface. On the other hand, placental and fetal EVs interact with endothelial cells, platelets, and immune cells, amplifying coagulation and inflammatory responses in the mother’s body.

In antiphospholipid syndrome, antiphospholipid antibodies (aPLs) may damage the mitochondria in the syncytiotrophoblast. One study has shown that extracellular vesicles (EVs) releasing mitochondrial DNA (mtDNA) following exposure to aPL significantly elevate the risk of pre-eclampsia by activating endothelial cells via the TLR-9 pathway [87].

Beyond coagulation, EVs also mediate immune interactions at the maternal–fetal interface. By transferring miRNAs and lncRNAs that regulate inflammatory cytokines and immune cell activation, EVs help establish immune tolerance and prevent inflammation-induced coagulation [88]. In contrast, in thrombophilic pregnancies, dysregulated EV content can lead to excessive immune responses, further exacerbating coagulation and compromising placental function [89,90].

## 5. Epigenetic Biomarkers as Diagnostic Tools for Thrombophilia-Related Complications

The stability of miRNAs, lncRNAs, and EVs in maternal blood makes them ideal candidates for non-invasive diagnostic testing in pregnancies affected by thrombophilia. By analyzing their levels and activity, clinicians can gain insights into the coagulation status and placental health of pregnant women with thrombophilia, potentially allowing for early intervention [91].

### 5.1. MiRNAs as Non-Invasive Diagnostic Biomarkers

The use of miRNAs as plasma biomarkers is very convenient due to their high stability, which is maintained even after exposure to extreme temperature conditions. However, there are some significant technical issues that need to be considered: sample extraction should be standardized, as miRNA contamination of plasma from red blood cells or from platelets could happen [92]

The role of miRNAs in modulating coagulation, endothelial function, and immune balance makes them highly valuable as biomarkers for thrombophilia-related pregnancy complications. Some specific miRNAs that influence key coagulation factors and endothelial integrity could be considered potential diagnostic biomarkers:miR-223: Known for its role in tissue factor regulation, miR-223 is associated with increased thrombin generation and a hypercoagulable state, making it a potential biomarker for assessing thrombotic risk related to APS, Factor V Leiden mutation, and protein C and S deficiencies [65].miR-210: Often upregulated under hypoxic conditions, miR-210 is involved in trophoblast invasion and placental development. Several studies have shown its involvement in processes associated with venous thrombosis, making it a good candidate as a biomarker for high-risk pregnancies associated with thrombophilias, especially with APS [56,93].miR-126: low levels of miR-126 are considered an indicator of endothelial dysfunction, being often observed in pre-eclampsia, making it a useful biomarker of severe maternal complications, including maternal death [94,95]. It is especially relevant for patients with APS and VTE.miR-145 and miR-19b: These miRNAs are involved in APS: they have elevated levels in patients with immune dysregulation and platelet activation, supporting their use as biomarkers for immune-mediated thrombosis and recurrent pregnancy loss [61,96,97].

### 5.2. LncRNAs as Emerging Diagnostic Biomarkers

LncRNAs offer complementary diagnostic insights by regulating coagulation, immune balance, and placental function. Certain lncRNAs have shown promise in diagnosing and monitoring pregnancy complications in thrombophilic patients:HOTAIR: Elevated in APS, HOTAIR is linked to immune regulation and inflammation, making it a potential biomarker for immune-related thrombotic complications, especially in patients with APS and Factor V Leiden mutation [78,79].MALAT1: Known for its role in promoting endothelial cell proliferation and angiogenesis, MALAT1 is downregulated in cases of pre-eclampsia and IUGR, indicating vascular insufficiency. Monitoring MALAT1 could offer insights into placental health and predict complications related to endothelial dysfunction [98].H19: This lncRNA is crucial for trophoblast invasion and placental development. Low H19 expression in maternal blood correlates with placental insufficiency, recurrent miscarriage, and pre-eclampsia, making it a valuable early marker of placental dysfunction in patients with congenital or acquired thrombophilias [99].

Combining lncRNA and miRNA analysis allows for a more comprehensive assessment of thrombophilia-related risks, enhancing diagnostic accuracy and supporting proactive clinical interventions.

### 5.3. Extracellular Vesicles as Diagnostic Biomarkers

The role of EVs as carriers of pro-coagulant factors, miRNAs, and lncRNAs offers additional diagnostic insights. EVs can reflect the physiological state of the placenta and maternal immune system, with the potential to indicate specific pregnancy risks:Tissue factor-bearing EVs: The presence of tissue factor on EVs contributes to a hypercoagulable state in thrombophilic pregnancies [100]. Elevated levels of these EVs in maternal blood correlate with a higher risk of placental thrombosis and vascular insufficiency, suggesting their utility as biomarkers for thrombotic complications [101,102].EVs as miRNA and lncRNA cargo: EVs carrying miRNAs and lncRNAs involved in coagulation and immune regulation can provide a non-invasive means of monitoring immune-mediated and vascular-related risks in pregnancy. For example, EVs enriched with miR-210 may reflect placental hypoxia, while EVs with high levels of HOTAIR could indicate immune dysregulation in APS [103].

Analyzing EV content offers a promising approach for developing comprehensive biomarker panels that combine miRNAs, lncRNAs, and EVs, providing a nuanced understanding of thrombophilia-related pregnancy risks.

## 6. Epigenetic Biomarkers for Therapy Assessment and Future Directions in Managing Thrombophilia-Related Pregnancy Complications

Clinical use of epigenetic biomarkers may represent a significant advance in the early diagnosis of complications associated with thrombophilia during pregnancy, as well as a new way of evaluating the effectiveness of therapeutic interventions (including anticoagulants) [104]. In the following section, we tried to integrate these new epigenetic markers into clinical management, as a necessary step for personalized therapy approaches.

### 6.1. The Role of Epigenetic Biomarkers in the Assessment of Anticoagulant Therapy Efficacy

Anticoagulant and antiplatelet drugs (including low molecular weight heparin—LMWH, and aspirin) are frequently prescribed to women diagnosed with high-risk forms of thrombophilia during pregnancy, but the therapeutic response is variable and inconsistent [105]. Current obstetric care lacks standardized, reliable markers to assess therapeutic efficacy, so MiRNAs, lncRNAs, and EVs may provide a real-time window into how well these therapies are working by reflecting dynamic changes in coagulation, immune response, and vascular health.

#### 6.1.1. MiRNAs as Therapeutic Biomarkers

MiRNAs represent interesting candidates for monitoring anticoagulant therapy efficacy. For instance, miR-223, which regulates tissue factor expression and thrombin generation, has been seen to diminish in patients treated with LMWH [106]. Its reduction is associated with fewer thrombotic events, indicating that miR-223 could serve as a biomarker for assessing anticoagulant efficacy during pregnancy in women with antiphospholipid syndrome (APS) or other thrombophilias [107,108]. Further studies should try to validate miR-223 as a biomarker across larger cohorts, potentially developing a non-invasive blood test to track the effectiveness of anticoagulant therapy.

Similarly, miR-210, which is dysregulated in hypoxia-related conditions such as pre-eclampsia, could be monitored to assess how well therapies are mitigating hypoxic stress in the placenta [109]. Changes in miR-210 levels could reflect improvements in placental oxygenation and trophoblast function, thereby serving as a predictive biomarker for the success of therapeutic interventions [110].

#### 6.1.2. LncRNAs and Therapeutic Monitoring

LncRNAs, though less explored in the context of therapy monitoring, could represent promising biomarkers for therapeutic response. For instance, lncRNA HOTAIR, which is involved in immune regulation and inflammation, has been linked to increased thrombotic risk in APS patients [79]. By tracking changes in HOTAIR levels during anticoagulant or anti-inflammatory therapy, clinicians could gain insights into the effectiveness of the treatment and make necessary adjustments [111,112].

#### 6.1.3. Extracellular Vesicles in Therapy Assessment

EVs, as carriers of pro-coagulant proteins, miRNAs, and lncRNAs, could also be used for the evaluation of therapy efficacy. For instance, EVs enriched with tissue factor are increased in response to thrombophilic conditions and reflect the overall coagulation status [100]. If the levels reduce after initiating anticoagulant therapy, it could indicate the therapy’s success in countering thrombosis. Moreover, monitoring EVs containing specific miRNAs and lncRNAs could offer a new way to evaluate the immune function and vascular health under therapy, making them useful in the context of a comprehensive therapeutic assessment [85,113] (Table 2).

### 6.2. Personalized Medicine: Tailoring Treatment Based on Biomarker Profiles

The variability in clinical outcomes among pregnant women with thrombophilia suggests that the current therapy approach is not satisfactory. Personalized medicine, tailoring treatments to individual molecular profiles, could improve outcomes significantly, with epigenetic markers offering the opportunity to refine risk stratification and develop more targeted therapeutic strategies.

#### 6.2.1. Risk Stratification Based on Biomarkers

One of the primary benefits of epigenetic biomarkers is their ability to stratify patients based on their risk for developing pregnancy complications. For instance, a combination of biomarkers like miR-126 (endothelial function), miR-210 (hypoxia and trophoblast invasion), and miR-223 (coagulation) could be used to assess the likelihood of developing pre-eclampsia, IUGR, or recurrent pregnancy loss. High-risk patients could then receive more intensive monitoring and earlier therapeutic interventions, while lower-risk patients might benefit from more conservative management.

#### 6.2.2. Targeting Epigenetic Mechanisms for Therapeutic Intervention

In addition to their diagnostic value, epigenetic markers like miRNAs and lncRNAs should be considered potential therapeutic targets [114]. For example, miRNA mimics or inhibitors could be used to correct dysregulated gene expression in thrombophilic pregnancies [115]. MiR-126, which is critical for maintaining endothelial health, could be targeted with a miRNA mimic to enhance angiogenesis and reduce endothelial dysfunction in conditions like pre-eclampsia [116]. Similarly, inhibiting miR-210 in pre-eclamptic women could improve trophoblast invasion and alleviate hypoxia, leading to better pregnancy outcomes [117].

On the lncRNA side, targeting molecules like HOTAIR, which regulates inflammatory responses in APS, could help reduce the immune activation that contributes to placental thrombosis [118]. Regulating the expression of lncRNAs involved in angiogenesis, such as MALAT1, could also enhance vascularization and reduce the risk of placental insufficiency in high-risk pregnancies [72].

Developing epigenetic-based therapies is still in the initial phase, but the potential benefits are significant. Future research should explore the safety and efficacy of targeting specific miRNAs and lncRNAs in pregnancy, with the goal of developing new therapeutic options for women with thrombophilia.

### 6.3. Use of Emerging Technologies for Biomarker Discovery and Validation

Recent advances in high-throughput sequencing and bioinformatics have revolutionized the field of epigenetic research, allowing for the discovery of novel miRNAs and lncRNAs involved in coagulation and pregnancy complications. These technologies have also made it possible to profile these markers in different body fluids, providing a non-invasive method for diagnosis and monitoring [119].

#### 6.3.1. Next-Generation Sequencing (NGS) and RNA Sequencing (RNA-seq)

Next-generation sequencing (NGS) and RNA-seq could facilitate the identification of novel miRNAs and lncRNAs associated with thrombophilia and pregnancy complications. These technologies allow for the simultaneous profiling of thousands of non-coding RNAs, providing a comprehensive view of the epigenetic landscape in thrombophilic pregnancies. By applying NGS to large patient cohorts, researchers can uncover new biomarkers that can predict negative pregnancy outcomes [120,121].

Incorporating bioinformatics tools and big data analytics into NGS studies could also help identify complex regulatory networks involving miRNAs, lncRNAs, and other molecular factors. For instance, machine learning algorithms could be used to analyze RNA-seq data and predict which miRNA or lncRNA signatures are associated with therapeutic response or disease progression. These insights may guide the creation of personalized therapy protocols for patients according to their distinct molecular profiles [122].

#### 6.3.2. Single-Cell Transcriptomics

Single-cell RNA sequencing (scRNA-seq) is an emerging technology that allows researchers to profile gene expression at the individual cell level. This approach is particularly valuable in studying the placenta and maternal–fetal interface, where various cell types (e.g., trophoblasts, endothelial cells, and immune cells) interact to regulate pregnancy outcomes [123,124]. By applying scRNA-seq to samples from thrombophilic pregnancies, researchers can identify cell-specific miRNAs and lncRNAs that are dysregulated. For example, scRNA-seq could reveal how miRNAs like miR-210 and lncRNAs like HOTAIR are expressed in individual trophoblast or immune cells, providing deeper insights into their role in placental development and immune regulation [125]. These findings may guide the creation of cell-targeted therapies that address the distinct pathophysiology of every type of thrombophilic pregnancy.

#### 6.3.3. Potential Utility of Artificial Intelligence

Artificial intelligence (AI) has emerged as a transformative tool in the discovery and interpretation of epigenetic biomarkers, offering significant potential in the diagnosis and prognosis of pregnancy-related complications of thrombophilias. AI algorithms, particularly those based on machine learning (ML) and deep learning (DL), can process vast datasets from high-throughput technologies such as RNA sequencing, microarrays, and proteomics to identify key miRNAs, lncRNAs, and extracellular vesicle-associated molecules associated with thrombophilic conditions [126,127]. These algorithms can detect complex patterns and interactions among biomarkers, providing insights into underlying molecular mechanisms and prioritizing candidates for further validation. In clinical applications, AI can enhance diagnostic accuracy by integrating epigenetic data with traditional biomarkers and clinical parameters, creating predictive models tailored to individual patient profiles [128]. AI-driven analytics also facilitate the identification of gestational age-specific reference ranges for biomarkers, improving their applicability during pregnancy. Additionally, AI tools can support therapeutic monitoring by correlating dynamic changes in biomarker expression with treatment response, enabling personalized approaches to manage thrombophilia-related complications [129]. Despite these advantages, challenges such as data standardization, interpretability of AI models, and integration into clinical workflows remain critical areas for future research and development [130].

### 6.4. Possible Study Limitations

The use of epigenetic biomarkers holds immense promise; however, there are several limitations to their large-scale clinical application. A major issue is the lack of standardized protocols for sample collection, biomarker extraction, and analysis. Differences in techniques for isolating miRNAs, lncRNAs, or EVs, as well as variations in pre-analytical handling, can lead to inconsistent results. Reproducibility across studies is limited, and no universally accepted reference ranges or cutoff values exist, particularly during pregnancy when biomarker levels naturally fluctuate [131].

The biological complexity of these biomarkers also presents challenges. miRNAs and lncRNAs often target multiple genes and pathways, making their effects on coagulation processes indirect and harder to interpret. EV populations are highly heterogeneous, and their isolation methods are labor-intensive and poorly standardized [132].

In clinical practice, integrating these novel biomarkers is hindered by the lack of regulatory approval, technological challenges, the need for specialized experts, and the high costs associated with advanced detection technologies like RT-qPCR and next-generation sequencing [133]. Current pregnancy-related diagnostic guidelines rely on established markers such as D-dimer and placental growth factor, making it challenging to incorporate novel tools without extensive validation [134]. Finally, while these biomarkers hold promise for early detection and personalized therapy, their clinical utility is limited by the absence of robust evidence correlating biomarker levels with specific pregnancy outcomes.

### 6.5. Future Directions in Clinical Translation and Biomarker Standardization

While research into epigenetic biomarkers has provided promising results, translating these findings into clinical practice remains a significant challenge. Several key steps need to be implemented to bridge the gap between research and real-world practice.

#### 6.5.1. Validation in Large and Diverse Populations

As most studies to date have been conducted in relatively small cohorts, often from homogenous populations, future research must focus on larger, multi-ethnic populations that reflect the diversity of thrombophilic patients, in order to enhance the generalizability of epigenetic biomarkers. This will allow for the development of reliable reference ranges and improve the accuracy of biomarker-based diagnostics [135].

#### 6.5.2. Standardization of Testing Protocols

Another critical barrier to the clinical use of epigenetic biomarkers is the lack of standardized testing protocols. While techniques such as qRT-PCR and RNA-seq are widely used in research, their application in clinical laboratories requires the development of high-throughput, cost-efficient assays that are both reproducible and precise. Standardization of these assays will guarantee the reliable measurement of miRNA and lncRNA levels across different laboratories and healthcare environments, facilitating their integration into routine clinical care [136].

#### 6.5.3. Incorporating Biomarkers into Clinical Decision-Making Algorithms

To fully integrate epigenetic biomarkers into clinical practice, clinicians will need tools to analyze biomarker data and implement it in patient care. One approach could be the development of risk calculators or decision-support algorithms that integrate epigenetic biomarkers with other clinical factors (e.g., family history, genetic predisposition, and previous pregnancy outcomes) to guide therapeutic decisions. These tools could help clinicians in tailoring specific clinical scenarios for individual patients, like pre-eclampsia screening, or monitoring anticoagulant therapy, ensuring that high-risk women receive the appropriate level of care to prevent pregnancy complications [137,138].

## 7. Conclusions

Thrombophilias and related coagulation disorders play a critical role in pregnancy complications, such as recurrent pregnancy loss, pre-eclampsia, and intra-uterine growth restriction (IUGR). These conditions disrupt key physiological processes, such as trophoblast invasion, angiogenesis, and immune regulation, which are essential for maintaining a healthy pregnancy. In recent years, research has highlighted the role of epigenetic regulation, particularly through microRNAs (miRNAs) and long non-coding RNAs (lncRNAs), in modulating these processes. These epigenetic factors have a big potential to serve as non-invasive biomarkers, offering real-time insights into the molecular and epigenetic changes that occur during pregnancy. MiRNAs detected in maternal blood, including miR-223, miR-210, and miR-126, have been identified as critical regulators of coagulation, angiogenesis, and immune responses, making them suitable for early diagnosis and therapeutic monitoring. Similarly, lncRNAs like HOTAIR and MALAT1 have emerged as important regulators of immune responses and endothelial function, confirming their utility as diagnostic tools.

The use of these biomarkers holds significant promise for personalized medicine. By stratifying patients based on their unique molecular profiles, clinicians can tailor treatment strategies to the individual needs of each patient. For instance, women at high risk for pre-eclampsia or IUGR could be identified early through miRNA or lncRNA profiling and undergo more intensive or targeted therapies, while lower-risk patients could be monitored more conservatively. This approach has the potential to improve outcomes by preventing the progression of pregnancy complications before clinical symptoms appear.

In conclusion, miRNAs, lncRNAs, and EVs represent an innovative approach to improving the diagnosis and management of thrombophilia-related pregnancy complications. Their integration into clinical practice could revolutionize the monitoring and treatment of these disorders, leading to better maternal and fetal outcomes. Future research should focus on validating these biomarkers, developing standardized testing methods, and incorporating epigenetic data into personalized treatment plans. By addressing these challenges, we can advance towards a future where thrombophilic pregnancies are handled with precision, minimizing risks to both mother and child.

## Figures and Tables

**Figure 1 ijms-25-13634-f001:**
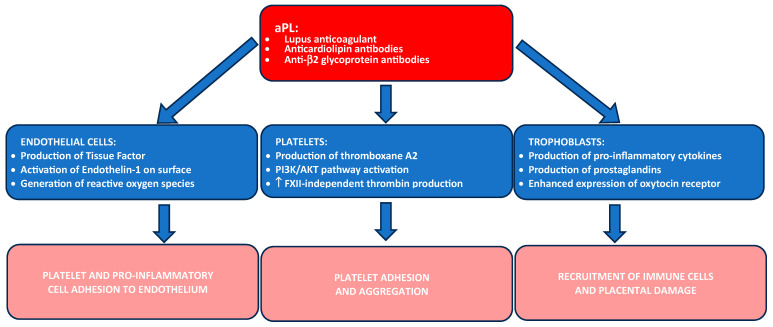
Key pathophysiological mechanisms of antiphospholipid syndrome during pregnancy (↑ = increase, PI3K = phopsphatidylinositol-3-kinase, AKT = protein kinase B).

**Figure 2 ijms-25-13634-f002:**
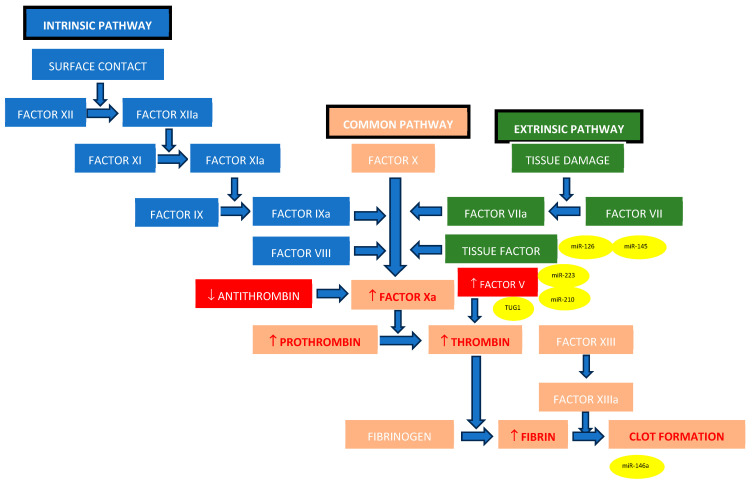
The most common modifications of the coagulation cascade induced by thrombophilias, including the key epigenetic biomarkers (modifications induced by thrombophilia are marked with red, while the epigenetic biomarkers are in yellow oval shapes; ↑ = elevation, ↓ = reduction).

**Table 1 ijms-25-13634-t001:** The impact of normal and impaired coagulation on the physiological processes during pregnancy.

	Pregnancy-Induced Normal Adaptations of the Coagulation System	Modifications Induced by Thrombophilias During Pregnancy
Adaptations in the coagulation cascade	- The maternal body shifts early during pregnancy to a hypercoagulable state, increasing clotting factors (fibrinogen, Factor VII, Factor VIII) to prevent hemorrhage during childbirth	- Congenital thrombophilias exaggerate this hypercoagulable state, causing excess thrombin production and a high risk of thromboembolic events - Acquired thrombophilias further increase clot formation, leading to placental vascular damage
Trophoblast formation and invasion	- Thrombin supports the invasion of the maternal endometrium by activating matrix metalloproteinases (MMPs), which remodel spiral arteries and establish the blood flow to the placenta	- In thrombophilias, excessive thrombin production can lead to premature fibrin deposition around spiral arteries, restricting the placental blood flow - In APS, antiphospholipid antibodies trigger trophoblast apoptosis, resulting in a disrupted blood supply and nutrient exchange
Angiogenesis and placental vascularization	- Adequate angiogenesis ensures placental development, supplying the fetus with oxygen and nutrients - Thrombin promotes endothelial cell proliferation and induces pro-angiogenic factors like VEGF for vessel formation	- Excessive thrombin activates anti-angiogenic pathways reducing VEGF expression and impairing vessel formation - Increased fibrin deposition further restricts blood flow and oxygenation
Immune regulation at maternal–fetal interface	- Immune tolerance is achieved through a balance between Th1 and Th2 responses, allowing the mother to tolerate the fetus while remaining protected from infections - Thrombin supports this immune modulation through PARs on immune cells	- Thrombophilias shift the immune balance toward a Th1-dominant pro-inflammatory response - Elevated thrombin levels promote pro-inflammatory cytokine release - The inflammatory response also disrupts immune tolerance
Nutrient and Oxygen Exchange	- Proper placental structure and blood flow ensure efficient nutrient and oxygen exchange between mother and fetus	- Thrombophilias can lead to excessive fibrin deposition in placental vessels, impairing nutrient and oxygen exchange

**Table 2 ijms-25-13634-t002:** Mechanisms of action, diagnostic potential, and therapeutic applications of epigenetic markers in thrombophilia-related pregnancy complications (LMWH = Low Molecular Weight Heparin).

Epigenetic Component	Description and Mechanism of Action	Role in Pregnancy and Coagulation	Diagnostic Potential	Therapy Assessment and Future Applications
miRNAs	Small non-coding RNAs that regulate gene expression by binding to mRNA, inhibiting translation or promoting degradation	**- Coagulation regulation:** Modulate key coagulation factors influencing thrombin production **- Vascular health:** Regulate endothelial function and angiogenesis **- Immune tolerance:** Balance pro- and anti-inflammatory signals	**- miR-223:** Marker for thrombin generation, assessing thrombotic risk. **- miR-210:** Hypoxia and placental development **- miR-126:** Early marker for endothelial dysfunction **- miR-145, miR-19b:** Markers for immune-mediated thrombosis in APS	**- miR-223:** Monitoring tool for anticoagulant efficacy, reducing thrombotic events in response to LMWH **- miR-210:** Evaluation of therapy impact on placental oxygenation **- Therapeutic targeting**: miRNA mimics/inhibitors could be explored to adjust miRNA levels
lncRNAs	Long non-coding RNAs that influence gene transcription, chromatin remodeling, and post-transcriptional regulation	**- Endothelial function:** Regulate vascular health, influencing VEGF production **- Coagulation control:** Affect coagulation factors like Factor V and VIII, modulating thrombin production **- Immune modulation:** Control pro-inflammatory cytokines, balancing immune responses	**- HOTAIR:** Elevated in APS, linked to immune response and thrombotic risk **- MALAT1:** Downregulated in pre-eclampsia/IUGR **- H19:** Low levels indicate placental dysfunction	**- HOTAIR:** Tracks immune and inflammatory response under therapy **- MALAT1:** Monitors endothelial health, assessing anti-inflammatory therapy **- Therapeutic targeting:** Target lncRNAs to reduce immune activation and thrombosis
Extracellular Vesicles (EVs)	Membrane-bound particles (exosomes and microvesicles) that transport proteins, miRNAs, and lncRNAs between cells	**- Coagulation:** Carry tissue factor and other pro-coagulant proteins, promoting the extrinsic pathway, increasing hypercoagulability in thrombophilic pregnancies **- Endothelial and immune modulation:** Transfer miRNAs/lncRNAs affecting vascular integrity and immune responses at the maternal–fetal interface	**- Tissue Factor-Bearing EVs:** Elevated levels indicate hypercoagulability, predictive of placental thrombosis and vascular insufficiency **- EV with miRNA/lncRNA cargo:** Presence of miR-210 (placental hypoxia) or HOTAIR (immune dysregulation) reflects specific pregnancy risks	**- EV Levels:** Used to assess anticoagulant therapy success **- Therapeutic cargo:** EVs could be engineered to carry therapeutic miRNAs/lncRNAs to target placental or vascular cells

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
