# Peer review of "Epigenetic Biomarkers in Thrombophilia-Related Pregnancy Complications: Mechanisms, Diagnostic Potential, and Therapeutic Implications: A Narrative Review"

_ijms, 2024, doi:10.3390/ijms252413634_

Round 1
Reviewer 1 Report
Comments and Suggestions for Authors
The paper is a very interesting account of thrombophilia during pregnancy, a well-known condition that causes numerous complications during pregnancy.
The treatment and study of pathophysiology based on epigenetics is something innovative and promising that really needs to be studied further. The text is interesting, but I didn't understand why deep vein thrombosis wasn't mentioned throughout the text, which is one of the complications caused by thrombophilia. Women in the gestational period have up to a six-fold increase in the risk of thrombosis compared to non-pregnant women.
(Bitsadze V, Khizroeva J, Alexander M, Elalamy I. Venous thrombosis risk factors in pregnant women. J Perinat Med. 2022 Jan 20;50(5):505-518. doi: 10.1515/jpm-2022-0008. PMID: 35044114.
James AH. Pregnancy and thrombotic risk. Crit Care Med. 2010 Feb;38(2 Suppl):S57-63. doi: 10.1097/CCM.0b013e3181c9e2bb. PMID: 20083915.)
Please justify in the answer why deep vein thrombosis was not mentioned or include this topic in the text.
The text presents several mechanisms, why was no diagram or figure made to summarize and facilitate the explanation? This would give the work greater visibility and citation.
Author Response
Thank you very much for taking the time to review this manuscript. Please find the detailed responses below and the corresponding revisions/corrections highlighted in red in the re-submitted files.
Comment 1: The treatment and study of pathophysiology based on epigenetics is something innovative and promising that really needs to be studied further. The text is interesting, but I didn't understand why deep vein thrombosis wasn't mentioned throughout the text, which is one of the complications caused by thrombophilia. Women in the gestational period have up to a six-fold increase in the risk of thrombosis compared to non-pregnant women. Please justify in the answer why deep vein thrombosis was not mentioned or include this topic in the text.
Response 1: Thank you for your remark, you are completely right. We have added a new subsection, 3.1. “Complications of thrombophilias during pregnancy”, which has a paragraph dealing with venous thromboembolism, deep venous thrombosis, and pulmonary embolism.
Comment 2: The text presents several mechanisms, why was no diagram or figure made to summarize and facilitate the explanation? This would give the work greater visibility and citation.
Response 2: Your suggestion was very helpful – we have included two diagrams, designed to facilitate the understanding of the text.
Reviewer 2 Report
Comments and Suggestions for Authors
authors should reduce the lenght of paragraphs concerning pathopsysiology of prothrombotic state during pregnancy and pregnancy related adverse outcomes.
furthermore, during the explanation of epigenetic factors associated to pregnancy management, they should specify that all dosable genetic molecules come from maternal side and not by foetal side in order to excape misunderstandings.
Author Response
Thank you very much for taking the time to review this manuscript. Please find the detailed responses below and the corresponding revisions/corrections highlighted in red in the re-submitted files.
Comment 1: Authors should reduce the length of paragraphs concerning pathophysiology of prothrombotic state during pregnancy and pregnancy related adverse outcomes.
Response 1: Thank you for your suggestion. We have tried to reduce the size of sections 2 and 3 as much as possible, trying to make them easier to understand.
Comment 2: Furthermore, during the explanation of epigenetic factors associated with pregnancy management, they should specify that all dosable genetic molecules come from maternal side and not from fetal side in order to escape misunderstandings.
Response 2: We have included a relevant comment at the beginning of section 4.
Reviewer 3 Report
Comments and Suggestions for Authors
Thank you for the invitation to review this work. The paper provides a narrative review of the role of epigenetic biomarkers (miRNAs, lncRNAs, and extracellular vesicles) in thrombophilia-related pregnancy complications, highlighting their diagnostic and therapeutic potential in personalizing obstetric care. However, there are several comments for authors to consider:
- While the authors propose the utility of epigenetic biomarkers, their diagnostic and therapeutic applications remain speculative without robust experimental or clinical validation.
- The manuscript frequently speculates on the potential clinical applications of miRNAs, lncRNAs, and EVs without sufficient evidence to support these claims.
- For example, while miR-223 and HOTAIR are highlighted as promising biomarkers, the manuscript provides limited data on their sensitivity, specificity, or reliability in predicting pregnancy outcomes.
- The manuscript does not adequately address potential confounders in the clinical utility of epigenetic biomarkers, such as:
- Variability in miRNA/lncRNA expression due to genetic, environmental, or lifestyle factors.
- Differences in laboratory methods for biomarker detection (e.g., qRT-PCR, RNA-seq).
- The influence of anticoagulant or anti-inflammatory therapies on biomarker levels.
- The manuscript presents miRNAs and lncRNAs as broadly applicable to thrombophilia-related complications without distinguishing their roles in specific subtypes of thrombophilia (e.g., congenital vs. acquired thrombophilias like antiphospholipid syndrome).
- For instance, the role of miR-126 in endothelial function may be more relevant to pre-eclampsia than recurrent pregnancy loss, but this distinction is not clarified.
- While the manuscript mentions next-generation sequencing (NGS) and single-cell RNA sequencing (scRNA-seq), it does not critically evaluate their limitations, such as high costs, technical challenges, or the need for specialized expertise.
- The potential role of artificial intelligence (AI) in biomarker discovery and data interpretation is also overlooked [see Goh et al 2024 Annals NY Acad Sci]
- A detailed discussion on how specific coagulation factors interact with epigenetic regulators. For example:
- How do miRNAs like miR-126 and miR-210 modulate the expression of specific coagulation factors (e.g., Factor V, fibrinogen)?
- What is the role of EVs in delivering pro-coagulant molecules to the maternal-fetal interface? Please expand.
- How do miRNAs and IncRNAs compare with traditional diagnostic markers (e.g., D-dimer, placental growth factor) regarding accuracy and reproducibility?
- Suggest adding more targeted discussion for integrating epigenetic biomarkers into specific clinical scenarios (e.g., pre-eclampsia screening, monitoring anticoagulant therapy).
The manuscript needs significant editing for language and comprehension.
Author Response
Thank you very much for taking the time to review this manuscript. Please find the detailed responses below and the corresponding revisions/corrections highlighted in red in the re-submitted files.
Comment 1: While the authors propose the utility of epigenetic biomarkers, their diagnostic and therapeutic applications remain speculative without robust experimental or clinical validation.
The manuscript frequently speculates on the potential clinical applications of miRNAs, lncRNAs, and EVs without sufficient evidence to support these claims.
For example, while miR-223 and HOTAIR are highlighted as promising biomarkers, the manuscript provides limited data on their sensitivity, specificity, or reliability in predicting pregnancy outcomes.
Response 1: We have added several new studies and evidence to support our manuscript. However, please note that this area of biomarkers for predicting pregnancy outcomes of thrombophilias is still under initial evaluation, requiring significantly more research results in order to be ready for incorporation into the diagnosis and therapy algorithms. This was also the main reason why we were not able to write a systematic review. Nevertheless, we consider that the topic of our narrative review is worth to be considered and we have tried to improve its contents as much as possible, following your great suggestions.
Comment 2: The manuscript does not adequately address potential confounders in the clinical utility of epigenetic biomarkers, such as:
- Variability in miRNA/lncRNA expression due to genetic, environmental, or lifestyle factors.
- Differences in laboratory methods for biomarker detection (e.g., qRT-PCR, RNA-seq).
- The influence of anticoagulant or anti-inflammatory therapies on biomarker levels.
Response 2: We have included a section of study limitations at the end of the review, before the future directions, trying to summarize all the potential problems.
Comment 3: The manuscript presents miRNAs and lncRNAs as broadly applicable to thrombophilia-related complications without distinguishing their roles in specific subtypes of thrombophilia (e.g., congenital vs. acquired thrombophilias like antiphospholipid syndrome).
Response 3: We have tried to make the proper mentions for each of the analyzed biomarkers.
Comment 4: For instance, the role of miR-126 in endothelial function may be more relevant to pre-eclampsia than recurrent pregnancy loss, but this distinction is not clarified.
Response 4: We have added that mention regarding miR-126, together with a relevant explanation.
Comment 5: While the manuscript mentions next-generation sequencing (NGS) and single-cell RNA sequencing (scRNA-seq), it does not critically evaluate their limitations, such as high costs, technical challenges, or the need for specialized expertise.
Response 5: We have mentioned the limitations of the mentioned methods, thank you for the suggestion.
Comment 6: The potential role of artificial intelligence (AI) in biomarker discovery and data interpretation is also overlooked [see Goh et al 2024 Annals NY Acad Sci]
Response 6: We have included a more detailed discussion regarding the role of AI as suggested.
Comment 7: A detailed discussion on how specific coagulation factors interact with epigenetic regulators. For example: How do miRNAs like miR-126 and miR-210 modulate the expression of specific coagulation factors (e.g., Factor V, fibrinogen)?
Response 7: We have provided some detailed insights regarding the mentioned interactions between the coagulation factors and epigenetic regulators.
Comment 8: What is the role of EVs in delivering pro-coagulant molecules to the maternal-fetal interface? Please expand.
Response 8: We have provided an explanation in the relevant section of EVs.
Comment 9: How do miRNAs and IncRNAs compare with traditional diagnostic markers (e.g., D-dimer, placental growth factor) regarding accuracy and reproducibility?
Response 9: We have included a comparison in the limitations section.
Comment 10: Suggest adding more targeted discussion for integrating epigenetic biomarkers into specific clinical scenarios (e.g., pre-eclampsia screening, monitoring anticoagulant therapy).
Response 10: We have added your valuable mention into the relevant section (6.4.3).
Comment on the quality of English language: The manuscript needs significant editing for language and comprehension.
Response: We have performed extensive editing of the text, corrected bad grammar, and tried to improve understandability.
Round 2
Reviewer 1 Report
Comments and Suggestions for Authors
The paper entitled “Epigenetic Biomarkers in Thrombophilia-Related Pregnancy Complications: Mechanisms, Diagnostic Potential, and Therapeutic Implications. A Narrative Review”, has been carefully revised by the authors, an outline has been included to make it easier to understand and some points in the bibliography have been revised, it is more complete and I recommend publishing it in this format.
Reviewer 3 Report
Comments and Suggestions for Authors
No further comments